# Tussocks facilitate their neighbours mainly by ameliorating extreme temperatures in tropical high mountains

Fernando Pedraza[1], Diego García-Meza[1], Hugo Tovar[2], Carlos Martorell[1]*

1 Departamento de Ecología y Recursos Naturales, Facultad de Ciencias, Universidad Nacional Autónoma de México, Ciudad de México, Mexico, 2 Unidad de Servicios Bioinformáticos, Instituto Nacional de Medicina Genómica, Ciudad de México, Mexico

☯ These authors contributed equally to this work.
* martorell@ciencias.unam.mx

**Data Availability Statement:** All relevant data are within the paper and its Supporting information files.

## Abstract

Facilitation by tussocks is common in high-altitude tropical environments. It is thought that facilitation results from stress amelioration, but it is unclear which of the many stressors acting in these environments is ameliorated. We aimed at determining the relative importance of different stressors as drivers of facilitation by the tussock *Festuca tolucensis* in Mexico. We employed eight experimental treatments to manipulate five stressors in the field: minimum temperatures by using electric radiators that kept plants warm; maximum temperatures by means of reflective sand that precluded temperature build-up during the day; UV radiation by using screens opaque to UV; poor soil properties by comparing soils from beneath tussocks and from bare ground; and low water availability by adding vermiculite to the soil. The performance (survival and growth) of *Mexerion sarmentosum* (a plant usually associated with *Festuca*) in these treatments was compared to that recorded under tussocks and in bare ground. Amelioration of extreme temperatures had the largest positive effects on *Mexerion* survival. UV radiation and increased soil humidity did not affect survival, although humidity increased growth rates. Nevertheless, tussocks reduced the growth of *Mexerion*, which is consistent with observations of competition between plants and soil microorganisms favoured by tussocks. Our results highlight the importance of the extreme daily fluctuations in temperature that characterise tropical mountains as fundamental drivers of their dynamics.

## Introduction

High-altitude environments are harsh. UV radiation is very intense due to the comparatively thin atmospheric layer above mountains [1, 2]. In volcanoes, soils are frequently sandy, young and undeveloped, and thus may be deficient in nutrients and have a reduced water-holding capacity [2–4]. Moreover, in contrast with extratropical alpine environments, páramos (high-altitude tropical environments above the treeline) face extreme and rapid changes in temperature every day: freezing temperatures at night and very high soil-surface temperatures during

**Funding:** This work received support from Programa de Apoyos a Proyectos de Investigación e Innovación Tecnológica-Universidad Nacional Autónoma de México (https://dgapa.unam.mx/index.php/impulso-a-la-investigacion/papiit) in the form of grants (IN217607 and IN212618) awarded to CM.

**Competing interests:** The authors have declared that no competing interests exist.

the day are common throughout the year [3, 5, 6]. Thus, it is said that páramos experience summer every day and winter every night [7]. Plants living in páramos must face this suite of harsh and often rapidly changing stressors.

One way plants may cope with such hostile conditions is through interactions with neighboring individuals [8–10]. Some páramo plants ameliorate abiotic stress in their vicinity [11, 12] resulting in strong facilitation, i.e., a non-trophic interaction in which at least one species is favoured by the presence of another [8, 13]. There is evidence that high-altitude plant communities, such as páramos, are in fact one of the systems in which plant-plant facilitation is strong and most frequent worldwide [11].

Tussock grasses are common benefactors in páramos [5, 14–16]. In fact, tussocks have been considered to be ecosystem engineers [15], and are sometimes the most important facilitators in páramos [17]. Several studies from páramos throughout the world have found large numbers of species facilitated by tussocks [see 11 and references therein]. Tussocks are said to reduce the intensity of multiple stressors. They may buffer extreme temperatures and lessen UV incidence because they produce dense shade [4, 15, 18–21]. They also improve soil properties increasing fertility, reducing the proportion of sands in the soil, and producing organic matter, which results in increased humidity and water-holding capacity [11, 15, 21]. The intensity of facilitation has been shown to decrease from the canopy center outwards [21, 22], suggesting that the close spatial associations observed between species at high altitudes [13, 23] are related to stress reduction near the tussock. However, no studies have aimed to test how the protégés are affected by the environmental changes induced by the tussocks. We also ignore which of the multiple stressors that occur in páramos is most important in driving facilitation by tussocks. Given the importance of tussocks both in terms of their abundance in páramos and the large number of species associated to them, these questions deserve attention.

In this contribution, we test in the field five stress-amelioration mechanisms to determine which (if any) drive facilitation by the tussock *Festuca tolucensis*, and evaluate which are more important. Given the large variations in temperature in páramos [3, 5], we hypothesise the buffering of maximum and minimum temperatures to be a crucial driver of facilitation by *Festuca*. We analyse the effects of the amelioration of each of the five stressors on the survival and growth of recently-germinated individuals of *Mexerion sarmentosum*, a small rosette plant that is positively associated to *Festuca* at the study site [24]. Our results were compared with the performance of *Mexerion* growing under tussocks and in full-stress conditions on bare ground. If the amelioration of a given stressor operates in our system, we expect performance to improve compared to bare ground.

## Materials and methods

### Study site

The study was conducted at the Iztaccíhuatl volcano, Mexico, (19.12˚ N, 98.65˚ W), at 3980 m a.s.l. Climatic reports at Paso de Cortés, located 320 m below our study site, indicate a mean annual temperature of 5.5˚C and weak seasonality (NOAA n/d). However, temperature undergoes wide fluctuations throughout the day (mean annual maximum air temperature is 13.9˚C and the average minimum is -2.8˚C), and thus may act as an important stressor. The study site is dominated by the tussock *Festuca tolucensis* (Poaceae). Many species grow under these tussocks, out of the 24 species found at our study site, 63% showed significant positive association with *Festuca*. This trend is especially strong in rosette species, as 80% of them were positively associated to the tussock [24]. One of such species was *Mexerion sarmentosum* (Asteraceae), a perennial herb that remains a small (< 4 cm in diameter) rosette for much of its life cycle, but becomes a decumbent herb as large as 30 cm tall when reproductive. In our experiment we

used recently germinated plants with a mean diameter of 1.51 cm, and a height < 0.5 cm. In small individuals, all the leaves in the rosette are appressed to the ground.

## Stress amelioration experiment

On September 2008, we collected 500 recently-germinated *Mexerion* rosettes and transplanted them to peat pots (7 cm side, 8 cm tall) filled to the top with soil from bare ground where no other plants were growing (except when stated otherwise, see below) taken from the study site. These pots were chosen because, when buried in the ground, they allow the movement of water. As a result, the hydric potential of the soil in the pot matches that of the surrounding environment since the materials inside and outside the pot have similar hydraulic properties [25, 26]. The plants were kept for one month on the roof of a building located at Iztaccíhuatl Park at 3980 m a.s.l. where they were protected from the wind but exposed to direct sunlight. Plants were watered at least once a week. After one month, survivors were relocated in the field along with their pots. Pots were buried so the level of the soil in them matched that of the surrounding ground. Because pots were almost completely full, the portion of the pot that protruded from the soil was minimal, which promotes evaporation and reduces water availability [26]. Relocating the plant altogether with the pot also minimises transplant shock [27], which would have obscured our results. Pots may in principle preclude the roots of *Mexerion* from spreading and interacting with *Festuca*'s. There is some evidence for root competition for water between tussocks and herbs in arid environments [28]. Nevertheless, our *Mexerion* individuals were very small and it seems unlikely that their roots would have spread beyond the limits imposed by the pot even if it were absent. Plants were randomly assigned to eight treatments designed to test the effect of different stressors and analyse the mechanisms underlying facilitation by tussocks:

**Tussock treatment (TUS).** *Mexerion* individuals grown in pots with soil from beneath tussocks were placed under a tussock canopy >80 cm in diameter. In this positive control all the stressors analysed in this study are expected to be alleviated.

**Fully exposed treatment (EXP).** As in the remainder of the treatments, plants were placed in areas without vegetation present in a radius of at least 1.5 m. The soil used was that of the study site where no other vegetation occurred in the vicinity. *Mexerion* in this negative control experienced all the stressors analysed in this study.

**Soil conditions treatment (SOI).** As EXP, but pots were filled with soil from beneath tussocks prior to transplant. This soil was expected to have more nutrients and greater water-holding capacity than that from bare areas [11, 29]. This procedure allows an evaluation of the effects that tussocks have by changing soil properties, but it does not provide direct evidence of which properties are changed (nutrients, soil texture, organic matter, etc.).

**Reduced hydric stress treatment (HYD).** As EXP, but pots contained a mixture of three-parts soil per one of vermiculite before the *Mexerion* seedlings were transplanted. Vermiculite absorbs large amounts of water and releases it gradually to the soil [30, 31]. Thus, the plants in this treatment were expected to have access to water for longer periods, as it may happen under tussocks. Results from this treatment must be interpreted carefully, as vermiculite may also affect pH and nutrient availability after some time in the soil [31].

**Minimum temperature amelioration treatment (MIN).** Temperature under tussock during the night has been reported to be higher than in bare areas (Coe 1969; Hedberg & Hedberg 1979). To keep *Mexerion* warm during this low-temperature time of the day we used electric radiators. These consisted of a 700 W, 120 V electric resistance contained in $15 \times 4$ cm metal cases and connected to a thermostat that could be regulated. This equipment was designed and manufactured specially for the experiment by KinTel S.A. de C.V. (Mexico City,

Mexico). After some preliminary tests, we found that the best option to keep a relatively constant temperature throughout the night was to place the radiator 5 cm away from the plants. We then regulated the thermostat so that the mean night temperature 5 cm away from the radiator equalled that recorded under tussocks. The measurements of temperature and calibration of the radiators were conducted in December.

**Maximum temperature amelioration treatment (MAX).** Tussocks also keep the daytime temperatures milder than those in bare areas [18, 20]. In high mountains, soil surface temperatures are high enough to be detrimental to plants [4]. As a way of lowering soil temperatures near the soil surface without interfering with photosynthetically active radiation, we covered the soil with a thin layer (~2 mm) of marble sand. This product is white, so we expected it to reduce soil temperature during the day by increasing the albedo. The diameter of the particles was similar to that of the sandy soils at the study site in order to minimise differences in texture that could affect water movement. Plants were placed at the centre of a 0.25 m$^2$ square covered with white marble sand.

**No ultraviolet radiation treatment (UV-).** We set $1 \times 1$ m Mylar screens 0.4 m above ground. The space between the soil and the screen was left open to allow the movement of air and minimize the screen's effect on temperature and air humidity. Mylar is opaque to UV radiation below 0.314 μm (UV-B), but is transparent to the rest of the spectrum [32]. The screens were perforated in a 5 cm grid to allow rainfall to pour in. This treatment intended to resemble the reduction in UV radiation caused by tussocks.

**Ultraviolet radiation treatment (UV+).** The presence of a screen in the UV- treatment may affect temperature, wind, humidity and precipitation, which in turn can affect *Mexerion* performance. This could result in confounded effects that would make it impossible to attribute the effects of the screen to UV reduction *per se*. As a control for the UV- treatment, we used Tedlar screens which are optically similar to Mylar but do not interfere with UV radiation [33, 34]. UV-opaque screens also affect several environmental variables, but they do so in ways identical to UV-transparent screens. Thus, the differences between UV+ and UV- treatments can be safely ascribed to UV radiation [32, 34, 35].

We set the experiment in a 0.25 ha area near the Altzomoni high-mountain refuge, which was the highest place where electricity was available to power the MIN treatment. The study site seems pretty homogeneous spatially, so our data are likely to be representative of the overall conditions. We used a randomized complete-block design. Blocks were areas < 6 m in diameter, and that thus may experience similar conditions. In total we set 10 blocks, each having the eight treatments represented once. In each experimental unit (area affected by a screen, tussock, radiator, patch of marble sand, or spot on bare ground) we placed two plants (= two adjacent pots), serving as subsamples to increase the precision of our results [36]. Therefore, we have ten replicates (ten experimental units) in our experiment with two subsamples, meaning that 20 plants were subject to each treatment. The mean distance between each pair of pots was 1.10 m, and the mean distance between blocks was 9.24 m. We measured plant diameter (hereafter referred to as plant size) at the beginning of the experiments. Though many plant traits can be used as surrogates for performance (e.g. plant height, root growth or leaf traits), we chose to only measure diameter and longevity. Since *Mexerion* is a rosette that grows appressed to the ground in the shape of a flat circle, other size measurements such as height would have been uninformative. We visited the experimental site four times after the start of the experiments (at days 20, 93, 165 and 201 after plant transplant) to record plant size and survival. Six months after the start of the experiments, when only four *Mexerion* individuals were still alive we recorded the final size and survival measurements. We determined the longevity of all *Mexerion* individuals from the survival data recorded in the field.

## Measurements of environmental variables

We used HOBO Pro temp/external temp data loggers to monitor temperature every minute in TUS, MAX, MIN, UV+ and EXP from 12 December 2008 to 25 December 2008 and 28 April 2009 to 11 May 2009. These dates were selected because they correspond to the coldest (minimum temperature in December = -7˚C, the coldest of the year) and the warmest (minimum temperature in May = 0˚C, the second warmest) seasons of the year. We did not measure temperatures in UV- because it has been shown that there is no difference in temperature below screens that filter or transmit UV radiation [35]. We thus assumed that the temperature in UV- was identical to that in UV+. We have no data for the MAX treatment in April because coyotes chewed on the data-logger cables. We have two measurements for each treatment.

The data-loggers have two sensors: one for air temperature and a thermistor enclosed in a small metal pipe on the tip of a cable. Air temperature sensors and thermistors were placed immediately above the ground. Because the air temperature sensor is housed in a relatively large plastic case, measurements provide an averaged temperature representative of an area of about 30 cm$^2$. This is appropriate for most treatments, because preliminary measurements showed that there were only weak horizontal temperature gradients. In contrast, steep gradients were observed in MIN and MAX, so measuring temperatures slightly away from the plant would result in large errors. Thus, we used the thermistor in these two treatments because, due to their small size, thermistors allowed for measurements in the close vicinity of plants without interfering with them. For EXP we used both sensors, so we could compare the results of each. We found that the thermistor attains air temperature at night (difference between air sensor and thermistor $\approx 0.5$˚C), but gets several degrees ($> 10$˚C) above air temperature during daytime. This precludes a direct comparison between MIN, MAX and the other treatments during the day. To have daytime data that are comparable between MAX, EXP and TUS treatments (the ones expected to differ in temperature during daytime) we measured soil-surface temperature with a Fluke 62 mini infrared thermometer. These measurements were conducted in March 2017 using 15 replicates.

Minimum temperatures differed between treatments (December: $P < 0.001$, April: $P < 0.001$). MIN, UV+ and UV- increased temperature compared with EXP, whereas MAX had virtually no effect on nighttime temperatures (Fig 1). The same patterns were observed in April, with the exception of MIN, which did not differ from EXP (Fig 1) because temperatures did not drop low enough to activate the radiators. Maximum soil-surface temperatures in May differed greatly between TUS, MAX and EXP ($P < 0.001$). In bare soil, temperatures were even $> 75$˚C, while tussocks kept soil much cooler (Fig 1, compare EXP with TUS maximum temperature measurements). Marble sand prevented soil overheating, reaching temperatures only ~10˚C above those observed under tussocks (Fig 1).

The validity of the HYD treatment depends on vermiculite actually increasing soil hydric potential. We were unable to measure soil desiccation rates in the field due to unpredictable bad weather (sudden rainfall, heavily overcast days with nearly no evaporation) every time we tried to. Instead, we conducted an experiment filling five peat pots as the ones described above with soil collected from the study site, and five with a 3:1 soil:vermiculite mixture. In each pot we placed a Delmhorst GB-1 gypsum blocks, which allowed us to determine the water potential with a Delmhorst KS-D1 moisture meter. Water was added to all pots until the content was fully saturated. Pots were then allowed to drain in a dark room for 48 h, after which the soil water potential was near zero in both treatments. The pots were then placed in a greenhouse for 72 h, recording the water potential at ~8 h intervals. Soils from the study site lost moisture very rapidly in the greenhouse. After three days they were nearly dry. In contrast, soil

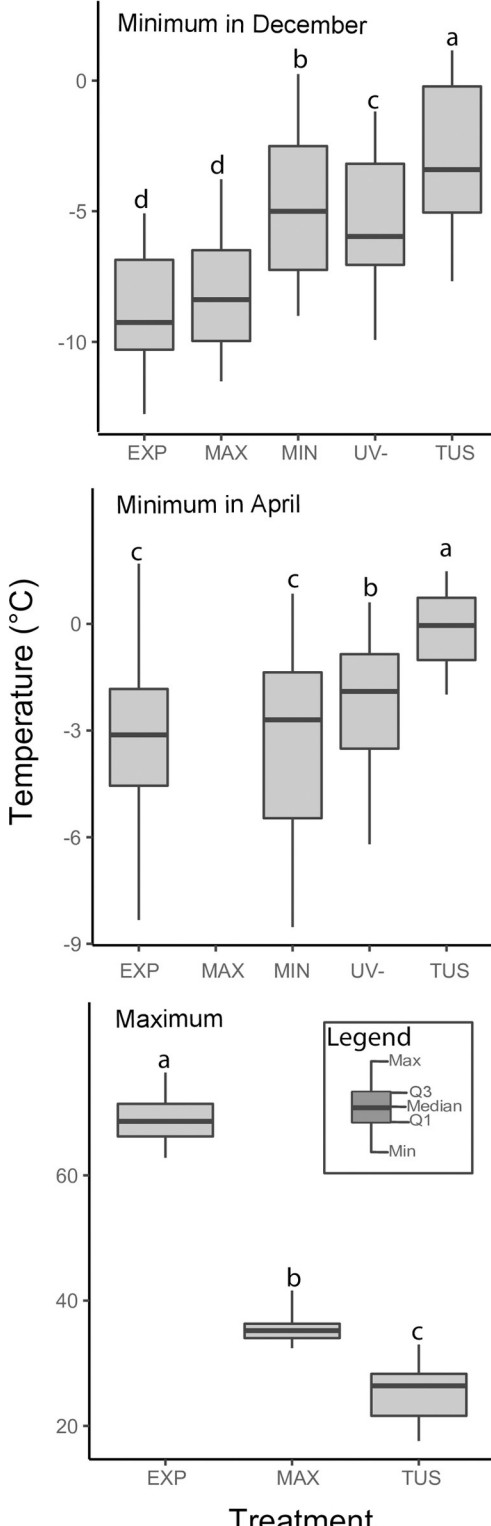

**Fig 1. Extreme temperatures recorded in the experiment.** Treatments are as follows: fully exposed treatment (EXP), maximum temperature amelioration treatment (MAX), minimum temperature amelioration treatment (MIN), no ultraviolet radiation treatment (UV-) and tussock treatment (TUS). We have no data for the MAX treatment in April because coyotes chewed on the thermistor. Shared letters indicate no significant differences ($\alpha$ = 0.05).

with vermiculite retained much of the water it had initially. After 72 h, soils with vermiculite had significantly less negative hydric potential ($P$ = 0.008, see Fig 1.1 in S1 Appendix).

## Statistical analyses

Plant longevity was defined as the number of days it survived. When an individual died between two observations, its longevity was recorded as the midway point between observations (Crawley, 2007). Plant life expectancy (i.e. mean longevity) was calculated using package 'survival' [37], by regressing each plant's longevity on initial plant size and treatment using a model with data censoring, a Weibull distribution and within-block variations accounted by a frailty term [38]. Package 'gamm4' [39] was used to analyse the change in plant size via a generalized additive mixed-effect model with Gaussian error [38]. For this model, fixed effects were: treatment, logged plant-size at the beginning of each observation period—*i.e.*, time between two consecutive size measurements—, and their interaction. The response variable was the logged size at the end of each observation period. Random components were individual plants nested in blocks crossed with the effect of time.

For all analyses significance was calculated from log-likelihood ratio tests. To determine differences between pairs of experimental treatments, we pooled all data for each possible pair of treatments and repeated the analysis to determine whether pooling caused a significant increase in unexplained deviance [38]. Minimum temperatures were analysed via mixed-effects linear models using 'lme4' [40] package for R [41]. Data logger and date were set as random crossed factors, and treatments as a fixed variable. Error was normal. For high temperatures an ANOVA was conducted because a single measurement was obtained from each experimental unit. Soil hydric potentials were compared by a Mann-Whitney $U$ test because of lack of normality.

## Results

Life expectancy differed between treatments ($P$ <0.001; Fig 2), increasing with their minimum temperatures (Spearman correlation between mean minimum temperature and life expectancy: $\rho$ = 0.71, $P$ = 0.048). The only prominent exception to this trend was MAX (Temperature-life expectancy correlation after removing MAX: $\rho$ = 0.95, $P$ < 0.001), which had a much larger survival than expected from its minimum temperature. Screens had positive effects on survival. However, because survival did not differ between UV+ and UV-, the increase in survival cannot be attributed to changes in UV radiation. Instead, this was likely the result of screens ameliorating minimum temperatures. SOI did not differ significantly from EXP.

Initial size had a strong effect on growth ($P$ <0.001), which was also affected by treatments ($P$ <0.001) but not their interaction. In most cases, plants shrank. Plants in HYD had the smallest reductions in size. However, HYD had also the lowest survival. In contrast, MIN caused the largest reductions in size, and did not differ from SOI and MAX (Fig 2). As before, no differences in growth were observed between treatments UV+ and UV-, indicating no effects from UV radiation (Fig 2).

## Discussion

Most of the treatments had some positive effects on *Mexerion*, although none was as effective as the tussock. However, given the extreme temperatures that we recorded in the field and the positive effect of temperature-ameliorating treatments, the regulation of extreme temperatures seems to be the most important factor. Growth rates revealed some negative effects caused by tussocks, which were seemingly related to soil conditions. In contrast, UV radiation had no

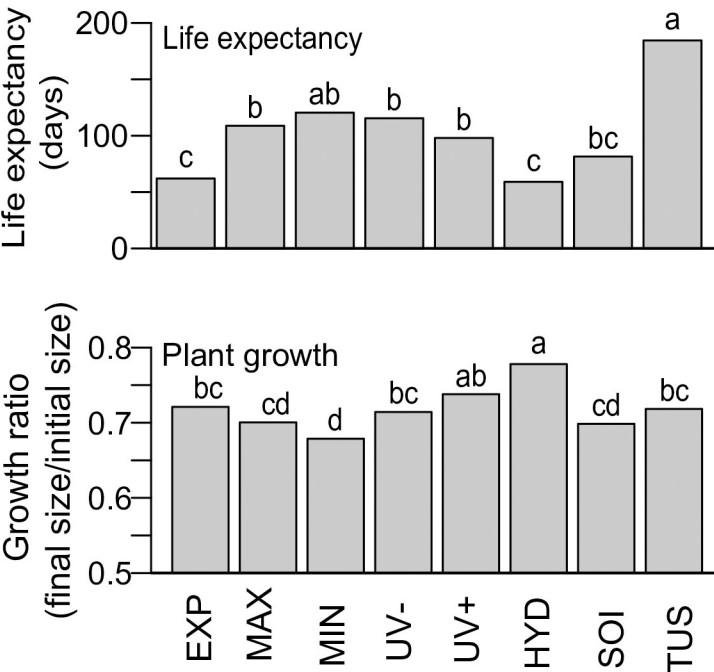

**Fig 2. Performance of *Mexerion* individuals in different treatments.** Treatments are as follows: fully exposed treatment (EXP), maximum temperature amelioration treatment (MAX), minimum temperature amelioration treatment (MIN), no ultraviolet radiation treatment (UV-), ultraviolet radiation treatment (UV+), reduced hydric stress treatment (HYD), soil conditions treatment (SOI) and tussock treatment (TUS). Life expectancy and plant growth correspond to mean-sized plants. Shared letters indicate no significant differences (α = 0.05).

effect on *Mexerion*'s performance, and the role of water in the plant-plant interaction was unclear.

As in other páramos [42], we recorded a large difference between daily minimum and maximum temperatures. The lowest minimum temperatures were recorded in EXP and the highest under TUS. Tussocks also had large effects on maximum temperatures, which were 40˚C lower than in EXP (Fig 1). Thus, tussocks are able to ameliorate both low and high temperatures, supporting the notion that they act as thermal buffers [6, 18, 19].

We found that plants under most treatments did not live as long as those in TUS, with those in EXP having the lowest level of life expectancy, highlighting the role of tussocks as facilitators. Factors related to the soil (HYD and SOI) had comparably low levels of life expectancy to EXP, indicating that they do not play an important role in determining life expectancy. The same can be concluded for UV radiation as plants in UV+ and UV- did not differ from each other in terms of both life expectancy and growth. However, it seems that increased water content in the soil promoted growth, suggesting that facilitation by tussocks may in part be due to water stress amelioration. The only treatments that caused a significant increase in life expectancy were those related with temperature, and MIN was the only treatment were plants lived as long as those in TUS. This indicates the important role that extreme temperatures play in facilitation by *Festuca*.

Life expectancy increased with minimum temperature (Fig 2). This indicates that extreme minimum-temperatures were a major driver of mortality in this high-altitude environment. Furthermore, the minimum temperatures (around -10˚C in December, and -4˚C in April) were close to the those expected to cause freezing damage to plants [43]. Therefore, the amelioration of minimum temperatures by tussocks appears to be a key driver of facilitation by

preventing freezing, as previously suggested [11]. This idea is supported by the fact that plants in MIN had the second largest life expectancy, and did not differ significantly from TUS.

A notable exception to the observed correlation between survival and minimum temperatures was MAX. This suggests that, unlike other treatments such as UV+ or UV-, increased minimum temperatures were not responsible of the relatively good performance of *Mexerion* individuals in the MAX treatment. This is what was expected. Whereas this treatment was incapable of reducing soil temperatures at noon as much as tussocks, it still caused a decrement of about 30°C in comparison with bare soil. The large positive effect of MAX on survival highlights the importance of maximum-temperature amelioration in páramos. While the effects of minimum temperatures on plants have been widely studied, maximum temperatures have been largely neglected in páramos. In our study, the maximum soil temperatures observed in EXP are high enough to cause irreversible damage to plant growth [44], whereas in TUS and MAX, temperatures may at most inactivate photosynthesis for short periods of time [45].

UV radiation is strongest at high altitudes near the equator [46]. Thus, plants living in páramos are expected to experience high levels of potentially lethal radiation. Excessive UV radiation has negative effects on plant life, damaging DNA, membranes and the photosynthetic apparatus [47]. In our experiment, screens had positive effects on plants, though this was not due to UV radiation, as survival and growth in UV+ and UV- treatments did not differ (Fig 2). Instead, the effect may be attributed to low temperature buffering under the screens.

The effects of water availability on *Mexerion* were unclear. Our results show that vermiculite increases water potential in the soil, as expected. Using models for soil desiccation based on soil temperature, it can be estimated that the difference between the hydric potentials in bare ground and under tussocks, increases at the same rate as that between soils with and without vermiculite (S1 Appendix). This suggests that the addition of vermiculite is an acceptable surrogate for the effects of tussocks. However, plants in HYD had the lowest survival. A lack of positive effects of vermiculite would be expected if moisture were not limiting. We consider that this is likely, because the removal of the topmost layers (~3–5 mm) of the soil revealed a very humid substrate during the first weeks of the experiment. Perhaps if *Mexerion* individuals had not died so rapidly in HYD, surviving into the drier months, positive effects of increased water availability on survival would have become apparent. Changes in soil chemistry due to the addition of vermiculite may also have obscured our data. Vermiculite tends to increase nutrient availability, more so if we consider that it neutralizes pH [31], and thus could be mobilising nutrients in the acidic soils (pH 5–6) of the Iztaccíhuatl volcano [48]. This would not account for the reduced survival in the HYD treatment, although it may explain why growth rates observed there were the highest.

Tussocks had not only positive, but negative effects on *Mexerion*, as evinced by the analyses of growth. Such negative effects may be caused by a reduction of photosynthetic radiation under the shade of *Festuca* [49], but also seem to be related to soil conditions and biota. Plants in SOI also had low growth rates in our experiment The use of soil from *Festuca* in SOI probably affected nutrients and soil biota, which is expected to be very abundant under tussocks [19]. In alpine environments, plants compete strongly for nutrients with soil microbes [50]. Nutrient-rich soils, such as those found under tussocks [19], favour microorganisms over plants, enhancing competition [50, 51] and ultimately leading to large reductions in plant growth [52, 53]. The idea that competition with microbes affects plants negatively is further supported by the fact that MIN and MAX had the most negative effects on *Mexerion* growth. Just as both treatments strongly promoted *Mexerion* survival, they may have favoured *Mexerion*'s microbial competitors by providing a more thermally-stable environment [54] and competition, leading to large reductions in plant size.

This study highlights the importance of considering the simultaneous effects of multiple stressors on facilitation. None of our treatments had positive effects as large as those observed under tussocks. This may in part be expected because our treatments were imperfect mimics of the ameliorating effects of tussocks. However, it would be surprising that, given the high intensity of different sources of stress, only one of them determines plant performance. Consider temperature: both extreme maximum and minimum temperatures had strong negative effects on plants growing on bare soil. However, tussocks ameliorate both of these adverse effects by maintaining protégées warmer during the night and cooler during the day. This joint effect may explain why plants in the TUS treatment had the largest observed life expectancies. The effects of tussocks on other factors may also contribute to making tussocks the most favourable environment in terms of survival. A full-factorial experiment would be required to analyse formally the joint effects of many stressors, but it would have been impossible to conduct (given 6 experimental forms of manipulation, we would have required $2^6 = 64$ experimental treatments).

Our results suggest a scenario in which the benefactor species exerts positive and negative, direct and indirect effects on its protégé through a multiplicity of environmental modifications. Such complex effects probably depend on the benefactor's identity: whereas facilitation by *Festuca* seemed independent of soil properties (although our data are not conclusive), these are important when cushion plants are considered [11, 55]. In turn, the protégé's tolerance to different stressors may determine its responses to the benefactor [56]. For instance, the negligible effect of UV radiation on *Mexerion* performance probably arises from its dense, reflective pubescence, which may confer resistance to UV radiation [47]. This interplay between amelioration of and tolerance to multiple stressors may explain why facilitative interactions are highly species-specific in nature [55, 57].

## Supporting information

**S1 Appendix. Comparison of the effects of vermiculite and tussocks on soil moisture.**
(PDF)

**S1 Data.**
(XLSX)

## Acknowledgments

D. Montañana assisted in the field. Parque Nacional Izta-Popo-Zoquiapan provided logistic support.

## Author Contributions

**Conceptualization:** Carlos Martorell.

**Data curation:** Hugo Tovar.

**Formal analysis:** Fernando Pedraza, Diego García-Meza.

**Funding acquisition:** Carlos Martorell.

**Methodology:** Diego García-Meza, Hugo Tovar, Carlos Martorell.

**Supervision:** Carlos Martorell.

**Writing – original draft:** Fernando Pedraza.

**Writing – review & editing:** Diego García-Meza, Hugo Tovar, Carlos Martorell.

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
