## [Decision Letter · Decision Letter 0]

26 Aug 2020

PONE-D-20-21043

Determining which mechanisms underlie facilitation by tussocks in tropical high mountains and their relative importance

PLOS ONE

Dear Dr. Martorell,

Thank you for submitting your manuscript to PLOS ONE. After careful consideration, we feel that it has merit but does not fully meet PLOS ONE’s publication criteria as it currently stands. Therefore, we invite you to submit a revised version of the manuscript that addresses the points raised during the review process.

We look forward to receiving your revised manuscript.

Kind regards,

Yajuan Zhu, Ph.D.

Academic Editor

PLOS ONE

Additional Editor Comments:

Two reviewers have given their questions and suggestions. Please answer these questions and modify your manuscript properly.

Journal Requirements:

Reviewers' comments:

Reviewer's Responses to Questions

**Comments to the Author**

1. Is the manuscript technically sound, and do the data support the conclusions?

Reviewer #1: Yes

Reviewer #2: Yes

2. Has the statistical analysis been performed appropriately and rigorously? 

Reviewer #1: Yes

Reviewer #2: Yes

3. Have the authors made all data underlying the findings in their manuscript fully available?

Reviewer #1: Yes

Reviewer #2: Yes

4. Is the manuscript presented in an intelligible fashion and written in standard English?

Reviewer #1: Yes

Reviewer #2: Yes

5. Review Comments to the Author

Reviewer #1: Interspecific relationships of plants is a core problem in plant ecology. In this study, authors split the facilitation by tussocks into effects of light, temperature, soil and water on plant survival and growth, so as to determine the relative importance of different stressors as drivers of facilitation by tussock Festuca tolucensis on Mexerion sarmentosum. I really like the rigorous and detailed experimental design. Although the paper is well written and structured, I still have a few concerns that I would like the authors to address. Below are my specific comments.

1) I don’t know why the authors only use two traits (plant longevity and plant size) to represent plant survival and growth under different treatments? Does plant size in the text mean plant diameter? There is no detail information in the M&M section. The effects of different pressures on plant growth are shown in different aspects, such as plant height, crown width, root growth, leaf traits, biomass, root/shoot ratio. Choosing the suitable traits to present the plant performance is very important in the study.

2) Reconsider what traits are included in the main paper figures vs. the supplemental. And I suggest moving the data on actual temperature treatments and hydric treatments achieved (results showed as Figure 1 and 2) to the methods section where it is first described. It’s just experimental treatment effects but not experiment results (except for the control with or without tussocks, which can be the experiment results).

3) “Five stressors in the field” (in the Abstract) and “plants were randomly assigned to eight treatments designed” (in the M&M section). Different expressions can easily mislead the reader.

4) In the discussion section, authors need to focus on the specific effect of different single stressors (UV- vs UV+, MIN vs EXP, MIN vs EXP, SOI vs EXP, HYD vs EXP) and the comprehensive effects of tussocks (TUS vs EXP) on plant performance.

Line 43: soils are frequently sandy in high-altitude environments? I don’t think so expanding to the globe.

Lines77-78: should be carefully.

Line 90: 5.5 oC

Lines 100-102: I suggest to delete this sentence.

Line 127: where did the soil in treatment EXP come from?

Line 186: “in four subsequent occasions” need more details.

Line 292-293: why it is the most important factor?

Appendix data: the last one is sheet “Max temperature” in the data description.

Reviewer #2: The authors try to find the mechanisms underlie facilitation by tussocks in tropical high mountains. The introduction is clear and sufficient, the experimental design is reasonable. And their results highlight the importance of the extreme daily fluctuations in temperature.

I just have a few suggestions.

1. The title should directly show the main conclusion.

2. In Fig.1 and Fig.3 legend. 'See text for treatment abbreviations.' Should list the full name of each treatment.

6. PLOS authors have the option to publish the peer review history of their article (what does this mean?). If published, this will include your full peer review and any attached files.

Reviewer #1: No

Reviewer #2: No

---

## [Author Response · Author response to Decision Letter 0]

10 Sep 2020

RESPONSE TO REVIEWER COMMENTS 

(Line numbers correspond to the document without change control)

REVIEWER 1

Comment 1. I don’t know why the authors only use two traits (plant longevity and plant size) to represent plant survival and growth under different treatments? Does plant size in the text mean plant diameter? There is no detail information in the M&M section. The effects of different pressures on plant growth are shown in different aspects, such as plant height, crown width, root growth, leaf traits, biomass, root/shoot ratio. Choosing the suitable traits to present the plant performance is very important in the study.

Answer: 

Yes, the influence of different stressors can be reflected on several plant traits. Our experimental design only considered the effect of stressors on two traits: plant size, measured as plant diameter, and plant longevity. Because we only measure two traits, our conclusions may be limited. Yet, we have two reasons for only measuring plant size and longevity. The first is to do with the plant we studied. Mexerion is a small rosette that occurs mainly appressed to the ground in the early stages of its life. This makes measuring plant height and root/shoot ratio difficult. We choose to measure plant diameter because it was an easy trait to measure. The second reason is to do with the experiment itself. Because we were also tracking plant longevity during the experiment, we did not want to measure a property during the experiment that could, in any way, have a negative effect on plants. As such, we did not want to unbury plants to measure root growth. We agree that measuring other properties that reflect plant performance would have strengthened our findings. Yet, we had to compromise given the constraints imposed by the studied plant and the experimental design, and thus only measured plant size and longevity. We have expanded the reasoning of our experimental design, it now read as (L185-190): ‘Though many plant traits can be used as surrogates for performance (e.g. plant height, root growth or leaf traits), we chose to only measure plant size and longevity because of two constraints. First, Mexerion is a rosette that grows appressed to the ground in the shape of a flat circle, making other size measurements such as height uninformative. Second, because we were tracking plant longevity, we could not use other traits that required an invasive measurement’

Plant size does in fact refer to plant diameter. We have amended our description in the M&M section, it now reads as (L184-5): ‘Plant diameter (hereafter referred to as plant size) and survival were measured at the beginning of the experiments.’

Comment 2. Reconsider what traits are included in the main paper figures vs. the supplemental. And I suggest moving the data on actual temperature treatments and hydric treatments achieved (results showed as Figure 1 and 2) to the methods section where it is first described. It’s just experimental treatment effects but not experiment results (except for the control with or without tussocks, which can be the experiment results).

Answer: 

We changed the data about temperature and hydric treatments and figure 1 to methods section (L222-236, 247-250). Figure 2 was sent to the appendix on soil moisture.

Comment 3. “Five stressors in the field” (in the Abstract) and “plants were randomly assigned to eight treatments designed” (in the M&M section). Different expressions can easily mislead the reader.

Answer: We have changed the wording in the abstract to clarify our statement. It now reads as (L24): ‘We employed eight experimental treatments to manipulate five stressors in the field’.

Comment 4. In the discussion section, authors need to focus on the specific effect of different single stressors (UV- vs UV+, MIN vs EXP, MIN vs EXP, SOI vs EXP, HYD vs EXP) and the comprehensive effects of tussocks (TUS vs EXP) on plant performance.

Answer: We agree with the point, the past version of the manuscript did not include a thorough comparison of individual treatments. We have now added a paragraph were these comparisons are included. Following this new paragraph, we maintained the previous structure of the discussion, focusing on particular stressors. The new paragraph reads as (L311-321): “We found that plants under most treatments did not live as long as those in TUS, with those in EXP having the lowest level of life expectancy, highlighting the role of tussocks as facilitators. Factors related to the soil (HYD and SOI) had comparably low levels of life expectancy to EXP, indicating that they do not play an important role in determining life expectancy. The same can be concluded for UV radiation as plants in UV+ and UV- did not differ from each other in terms of both life expectancy and growth. However, it seems that increased water content in the soil promoted growth, suggesting that facilitation by tussocks may in part be due to water stress amelioration. The only treatments that caused a significant increase in life expectancy were those related with temperature, and MIN was the only treatment were plants lived as long as those in TUS. This indicates the important role that extreme temperatures play in facilitation by Festuca.”

Comment Line 43: soils are frequently sandy in high-altitude environments? I don’t think so expanding to the globe.

Answer: True, soils may not be sandy in other high-altitude environments. We have rephrased the sentence to clarify that sandy soils are expected in high-altitude volcanic environments. It now reads as (L43): ‘High-altitude environments are harsh. UV radiation is very intense due to the comparatively thin atmospheric layer above mountains [1, 2]. In volcanoes, soils are frequently sandy…’

Comment Lines 77-78: should be carefully.

Answer: We have attenuated our claim. It now reads as (L78): ‘Given the large variations in temperature in páramos [3, 5], it seems likely that the buffering of maximum and minimum temperatures is an important driver of facilitation by Festuca’.

Comment Line 90: 5.5 oC

Answer: We have fixed our spelling mistake. (L90): 5.5 �C. 

Comment Lines 100-102: I suggest to delete this sentence.

Answer: We agree, the sentence did not bear particular relevance. We have deleted it. 

Comment Line 127: where did the soil in treatment EXP come from?

Answer: Soil for the EXP treatment came directly from the study site. Plants we directly planted in the study site in areas were no vegetation was present. We have rephrased our sentence to clarify this point. It now reads as (L126-7): ‘Fully exposed treatment (EXP): As in the remainder of the treatments, plants were placed in areas without vegetation present in a radius of at least 1.5 m. The soil used was that of the study site where no other vegetation occurred in the vicinity.’

Comment Line 186: “in four subsequent occasions” need more details.

Answer: We have now expanded our description of when we visited the experimental site to record measurements. It now read as (L190-193): ‘Plant diameter and survival were measured at the beginning of the experiments. We visited the experimental site four times after the start of the experiments (at days 20, 93, 165 and 201 after plant transplant) to record plant diameter and survival. Six months after the start of the experiments, when only four Mexerion individuals were still alive, we recorded the final size and survival measurements.’

Comment Line 292-293: why it is the most important factor?

Answer: Our temperature measurements in bare ground reveal very high (> 60 �C) and very low (< 0�C) temperatures to occur in our study site. Thus, we conclude that out of the environmental factors we screened, the most important individual one driving facilitation was extreme temperatures. We have reworded our sentence, it now reads as (L301-2): ‘However, given the extreme temperatures that we recorded in the field and the positive effect of temperature-ameliorating treatments, the regulation of extreme temperatures seems to be the most important factor.’

Comment Appendix data: the last one is sheet “Max temperature” in the data description.

Answer: We have corrected the data description in our appendix data file. It now correctly refers to the ‘Max temperature’ sheet. 

REVIEWER 2

Comment 1. The title should directly show the main conclusion.

Answer: We have updated our title. It now read as: ‘Tussocks facilitate their neighbours mainly by ameliorating extreme temperatures in tropical high mountains.’

Comment 2. In Fig.1 and Fig.3 legend. 'See text for treatment abbreviations.' Should list the full name of each treatment.

Answer: We have included the full name of each treatment for both figures 1 and 3 (note that the later is now fig 2).

---

## [Decision Letter · Decision Letter 1]

8 Oct 2020

PONE-D-20-21043R1

Tussocks facilitate their neighbours mainly by ameliorating extreme temperatures in tropical high mountains

PLOS ONE

Dear Dr. Martorell,

Thank you for submitting your manuscript to PLOS ONE. After careful consideration, we feel that it has merit but does not fully meet PLOS ONE’s publication criteria as it currently stands. Therefore, we invite you to submit a revised version of the manuscript that addresses the points raised during the review process.

Please answer the reviewer's question and upload a revised manuscript.

We look forward to receiving your revised manuscript.

Kind regards,

Yajuan Zhu, Ph.D.

Academic Editor

PLOS ONE

Additional Editor Comments (if provided):

Please answer the reviewer's question and upload revised manuscript.

Reviewers' comments:

Reviewer's Responses to Questions

**Comments to the Author**

1. If the authors have adequately addressed your comments raised in a previous round of review and you feel that this manuscript is now acceptable for publication, you may indicate that here to bypass the “Comments to the Author” section, enter your conflict of interest statement in the “Confidential to Editor” section, and submit your "Accept" recommendation.

Reviewer #1: All comments have been addressed

Reviewer #2: All comments have been addressed

2. Is the manuscript technically sound, and do the data support the conclusions?

Reviewer #1: Yes

Reviewer #2: Yes

3. Has the statistical analysis been performed appropriately and rigorously? 

Reviewer #1: Yes

Reviewer #2: Yes

4. Have the authors made all data underlying the findings in their manuscript fully available?

Reviewer #1: Yes

Reviewer #2: Yes

5. Is the manuscript presented in an intelligible fashion and written in standard English?

Reviewer #1: Yes

Reviewer #2: Yes

6. Review Comments to the Author

Reviewer #1: The hypothesis posited to be a little naive in that it is very likely that all five factors are going to have an effect on the plant performance. i'd like a bolder hypothesis, based (obviously) on theory, that which factor (or factors) will be more preponderant than the others.

Line 185: how did author measure plant survival at the beginning of the experiments? And the author need to indicate that the measure of survival is for the measure of longevity.

Lines 189-190: the second reason is strained. For example, you can harvest the plant for biomass at the end of the experiment. So I suggests to delete it.

Reviewer #2: The authors have adequately addressed the comments raised in a previous round of review and I feel that this manuscript is now acceptable for publication.

7. PLOS authors have the option to publish the peer review history of their article (what does this mean?). If published, this will include your full peer review and any attached files.

Reviewer #1: No

Reviewer #2: No

---

## [Author Response · Author response to Decision Letter 1]

8 Oct 2020

RESPONSE TO REVIEWER COMMENTS

(Line numbers correspond to the document without change control)

REVIEWER 1

Comment 1: The hypothesis posited to be a little naive in that it is very likely that all five factors are going to have an effect on the plant performance. i'd like a bolder hypothesis, based (obviously) on theory, that which factor (or factors) will be more preponderant than the others.

Answer: We agree with the comment, the hypothesis was in fact too vague. We have rewritten it to highlight the expected role of extreme temperature buffering in driving facilitation in páramos. It now reads as (L74-76): ‘Given the large variations in temperature in páramos [3, 5], we hypothesise the buffering of maximum and minimum temperatures to be a crucial driver of facilitation by Festuca.’

Comment 2, Line 185: how did author measure plant survival at the beginning of the experiments? And the author need to indicate that the measure of survival is for the measure of longevity.

Answer: Our description of survival and longevity was not clear. We have modified the sentence which read ‘We measured plant diameter (hereafter referred to as plant size) and survival were measured at the beginning of the experiments’. It now reads as (L188-189): ‘We measured plant diameter (hereafter referred to as plant size) at the beginning of the experiments’. Furthermore, to make clear how we used the survival data, we added the following sentence (L196-197): We determined the longevity of all Mexerion individuals from the survival data recorded in the field. 

Comment 3, Lines 185-190: the second reason is strained. For example, you can harvest the plant for biomass at the end of the experiment. So I suggests to delete it.

Answer: We have deleted the sentence. The string of sentences now read as (188-193): ‘We measured plant diameter (hereafter referred to as plant size) at the beginning of the experiments. Though many plant traits can be used as surrogates for performance (e.g. plant height, root growth or leaf traits), we chose to only measure diameter and longevity. Since Mexerion is a rosette that grows appressed to the ground in the shape of a flat circle, other size measurements such as height would have been uninformative.’

---

## [Decision Letter · Decision Letter 2]

2 Nov 2020

Tussocks facilitate their neighbours mainly by ameliorating extreme temperatures in tropical high mountains

PONE-D-20-21043R2

Dear Dr. Martorell,

We’re pleased to inform you that your manuscript has been judged scientifically suitable for publication and will be formally accepted for publication once it meets all outstanding technical requirements.

Kind regards,

Yajuan Zhu, Ph.D.

Academic Editor

PLOS ONE

Additional Editor Comments (optional):

The authors had answered reviewer's questions and improved the manuscript. Now it's ready for publication.

Reviewers' comments:

Reviewer's Responses to Questions

**Comments to the Author**

1. If the authors have adequately addressed your comments raised in a previous round of review and you feel that this manuscript is now acceptable for publication, you may indicate that here to bypass the “Comments to the Author” section, enter your conflict of interest statement in the “Confidential to Editor” section, and submit your "Accept" recommendation.

Reviewer #1: All comments have been addressed

2. Is the manuscript technically sound, and do the data support the conclusions?

Reviewer #1: (No Response)

3. Has the statistical analysis been performed appropriately and rigorously? 

Reviewer #1: (No Response)

4. Have the authors made all data underlying the findings in their manuscript fully available?

Reviewer #1: (No Response)

5. Is the manuscript presented in an intelligible fashion and written in standard English?

Reviewer #1: (No Response)

6. Review Comments to the Author

Reviewer #1: (No Response)

7. PLOS authors have the option to publish the peer review history of their article (what does this mean?). If published, this will include your full peer review and any attached files.

Reviewer #1: No

---

## [Editor Report · Acceptance letter]

6 Nov 2020

PONE-D-20-21043R2 

Tussocks facilitate their neighbours mainly by ameliorating extreme temperatures in tropical high mountains. 

Dear Dr. Martorell:

I'm pleased to inform you that your manuscript has been deemed suitable for publication in PLOS ONE. Congratulations! Your manuscript is now with our production department. 

Kind regards, 

on behalf of

Dr. Yajuan Zhu 

Academic Editor

PLOS ONE